# Synergistic Interaction of the Class IIa HDAC Inhibitor CHDI0039 with Bortezomib in Head and Neck Cancer Cells

**DOI:** 10.3390/ijms24065553

**Published:** 2023-03-14

**Authors:** Christian Schrenk, Lukas M. Bollmann, Corinna Haist, Arthur Bister, Constanze Wiek, Maria Wecker, Dennis Roth, Patrick Petzsch, Karl Köhrer, Alexandra Hamacher, Helmut Hanenberg, Georg Fluegen, Matthias U. Kassack

**Affiliations:** 1Institute of Pharmaceutical and Medicinal Chemistry, Heinrich-Heine-University Duesseldorf, 40225 Duesseldorf, Germany; 2Department of Otorhinolaryngology, Head & Neck Surgery, Heinrich-Heine-University Duesseldorf, 40225 Duesseldorf, Germany; 3Department of General, Visceral and Pediatric Surgery, Medical Faculty, University Hospital of the Heinrich-Heine-University Duesseldorf, 40225 Duesseldorf, Germany; 4Biological and Medical Research Centre (BMFZ), Genomics & Transcriptomics Laboratory, Heinrich-Heine-University Duesseldorf, 40225 Duesseldorf, Germany; 5Department of Pediatrics III, University Children’s Hospital Essen, University of Duisburg-Essen, 45147 Essen, Germany

**Keywords:** class IIa histone deacetylase, HDAC4, HDAC5, HDAC inhibitor, CHDI0039, bortezomib, head and neck cancer

## Abstract

In contrast to class I/IIb/pan histone deacetylase inhibitors (HDACi), the role of class IIa HDACi as anti-cancer chemosensitizing agents is less well understood. Here, we studied the effects of HDAC4 in particular and the class IIa HDACi CHDI0039 on proliferation and chemosensitivity in Cal27 and cisplatin-resistant Cal27CisR head and neck squamous cell cancer (HNSCC). HDAC4 and HDAC5 overexpression clones were generated. HDAC4 overexpression (Cal27_HDAC4) increased proliferation significantly compared to vector control cells (Cal27_VC). Chicken chorioallantoic membrane (CAM) studies confirmed the in vitro results: Cal27_HDAC4 tumors were slightly larger than tumors from Cal27_VC, and treatment with CHDI0039 resulted in a significant decrease in tumor size and weight of Cal27_HDAC4 but not Cal27_VC. Unlike class I/pan-HDACi, treatment with CHDI0039 had only a marginal impact on cisplatin cytotoxicity irrespective of HDAC4 and HDAC5 expression. In contrast, the combination of CHDI0039 with bortezomib was synergistic (Chou–Talalay) in MTT and caspase 3/7 activation experiments. RNAseq indicated that treatment with CHDI0039 alters the expression of genes whose up- or downregulation is associated with increased survival in HNSCC patients according to Kaplan–Meier data. We conclude that the combination of class IIa HDACi with proteasome inhibitors constitutes an effective treatment option for HNSCC, particularly for platinum-resistant cancers.

## 1. Introduction

Genetic and epigenetic alterations are involved in the development, progression and also (chemo) resistance of cancer [1]. Acetylation of nuclear histone proteins is an extensively studied posttranslational modification, essential for DNA replication and gene expression, that is tightly regulated by the opposing effects of histone acetyltransferases (HATs) and histone deacetylases (HDACs). In humans, 18 HDAC enzymes have been identified. The zinc-dependent HDAC enzymes comprise class I (HDAC1-3, HDAC8), class IIa (HDAC4, 5, 7, 9), class IIb (HDAC6 and 10), and class IV (HDAC11) [2,3,4]. Members of class IIa HDACs play an important role in a variety of human malignancies, e.g., multiple myeloma, glioma, leukemia and also pancreatic, urothelial, ovarian, colon and gastric cancer [1,3,4,5,6,7,8,9,10,11,12,13]. The oncogenic potential of class IIa HDACs has been shown by their repressive function on tumor suppressor genes, such as p21 and p27, thereby affecting the cell cycle and cell proliferation [12]. Class IIa HDACs are also involved in the programmed cell death, e.g., HDAC4 knockdown in colon cancer induced apoptosis, which was associated with increased p21 expression [13]. Furthermore, members of class IIa HDACs, such as HDAC4 and HDAC7, display transforming capabilities [14]. While this pro-cancer role of class IIa HDACs has primarily been studied in overexpression and knockdown strategies in vivo and in vitro models, it remains unclear whether chemical inhibition is sufficient to reverse the pro-cancer function/impact of class IIa HDACs. 

Several studies and clinical trials have been centered around the anti-cancer activity of class I or pan-HDACi. In fact, all FDA-approved HDACi inhibit class I HDACs, among other classes [15]. These approved HDACi can induce cell cycle arrest and apoptosis [16]. At the same time, they have a plethora of side effects that can limit the therapeutic use in patients [17]. One of the main mechanisms of action of HDACi is thought to be the regulation of histone modification, mainly caused by the inhibition of class I HDACs. In comparison to class I and pan-HDACi, the anti-cancer effects of class IIa HDACi have received considerably less attention in the literature. Reasons might be the highly reduced deacetylase activity compared to class I HDACs and also the limited number of selective and potent class IIa HDACi that had been available. Moreover, some class IIa HDACs can exert their repressive function on target genes independently of their deacetylase domain (e.g., MITR, a splice variant of HDAC9 that completely lacks the deacetylase domain). Ultimately, class IIa HDACi are hypothesized to be less powerful than class I and pan-HDACi [18]. However, using selective class IIa HDACi might provide an advantage in terms of their lower overall cytotoxicity and their anti-cancer effects in specific cancer subtypes [16,19]. 

Overcoming intrinsic or acquired drug resistance is a major challenge to improve the outcome/prognosis of cancer patients and to prevent cancer recurrence and dissemination. Synergistic interaction of HDACi with several established anti-cancer drugs, such as platinum compounds, temozolomide, bicalutamide, proteasome, and kinase inhibitors, highlights the huge potential of epigenetic modulation in cancer treatment [20]. Recently, class IIa HDACi have emerged as potential anti-cancer drugs that synergize with proteasome inhibitors by regulating the expression and activity of transcription factors. In multiple myeloma (MM), HDAC4 protected cells from ER-stress-mediated apoptosis by repressing activating transcription factor 4 (ATF4). ATF4 expression in the nucleus was increased upon knockdown or inhibition of HDAC4 under ER-stress conditions, which was associated with increased proapoptotic CHOP (C/EBP homologous protein) transcription factor expression enhancing cytotoxicity in MM cells [9]. In another study, the selective class IIa HDACi TMP269 showed transcription-dependent anti-tumor effects in pancreatic cancer by upregulation of the transcription factor FOXO3a, which led to inhibited cell growth and G1/S arrest in AsPC-1 cells [10]. Since FOXO3a expression is regulated by proteasomal degradation, the combination of TMP269 with the proteasome inhibitor carfilzomib drastically inhibited cell growth [9]. In leukemia cell lines, we have recently reported about a synergistic effect of the novel class IIa HDACi YAK540 in combination with bortezomib [21]. Other studies in gastric cancer have shown that HDAC4 inhibition can also increase cisplatin and docetaxel cytotoxicity [7].

The purpose of this study was to investigate the role of HDAC4 (and HDAC5) in the head and neck squamous cancer cell (HNSCC) line Cal27 and its cisplatin-resistant subline Cal27CisR. This cell pair has extensively been studied by our group to understand mechanisms of cisplatin resistance and to develop strategies to overcome this chemoresistance, including the use of pan-HDACi and class I HDACi [16,18,22,23,24]. As increased expression of HDAC4 in HNSCC compared to normal tissue has been reported [25,26], we were encouraged to study the role of class IIa HDACs, in particular HDAC4, in our cisplatin-sensitive and -resistant HNSCC pair. In general, class IIa HDACs are expressed at lower levels compared to class I HDACs. Therefore, we established HDAC4 and HDAC5 overexpression clones using a lentiviral system to investigate class IIa HDAC effects and their pharmacological inhibition by the selective and potent class IIa HDACi CHDI0039 [27]. We were able to demonstrate that class IIa HDAC inhibition reduces the growth of cancer cells with increased HDAC4 expression in vitro and in vivo (CAM model). Furthermore, CHDI0039 treatment synergistically enhances bortezomib-mediated cytotoxicity via increased caspase 3/7 activation.

## 2. Results

### 2.1. Establishment of a Cellular Test System to Study the Role of Class IIa HDACs in Head and Neck Cancer

To investigate the role of class IIa HDACs in head and neck cancer and possible implications for chemoresistance, we generated stable HDAC4 and HDAC5 overexpression clones by lentiviral transduction. The head and neck carcinoma cell line Cal27 and the cisplatin-resistant subline Cal27CisR were used as a model system for chemosensitivity/chemoresistance. First, expression of the zinc-dependent HDACs was analyzed. Gene expression analysis (RNA sequencing) was performed with cell lines Cal27_VC (vector control) and Cal27_HDAC4 (HDAC4 overexpression clone) (Figure 1a). In Cal27_VC cells, class I HDACs displayed a much higher expression than HDAC4 and 5 (Figure 1a, white bars), whereas in Cal27_HDAC4 cells (Figure 1a, black bars), HDAC4 expression was massively upregulated (54-fold) to the level of HDAC1. All other HDAC isoforms in Cal27_HDAC4 showed similar expression as in Cal27_VC. Western blot analysis of HDAC isoenzymes was performed in all cell lines (Cal27_VC, Cal27_HDAC4, Cal27_HDAC5, Cal27CisR_VC, Cal27_HDAC4, and Cal27_HDAC5) and confirmed overexpression of HDAC4 and HDAC5 in the respective HDAC4/HDAC5 overexpression clones (Figure 1b). Interestingly, the chemoresistant subline Cal27CisR showed significantly elevated protein expression of HDAC2 (class I), and HDAC11 (class IV) compared to the parental cell line Cal27, confirming previous studies of our group on increased effectivity of class I HDAC inhibitors in chemoresistant cell lines [18]. Elevated protein expression is confirmed by quantification of the Western blot data and is shown in Appendix A. The cisplatin-resistant cell line Cal27CisR is about 4-fold more resistant to cisplatin compared to Cal27, and was generated by an intermittent treatment of Cal27 with cisplatin over several weeks (Figure 1c) [22].

**Figure 1 ijms-24-05553-f001:**
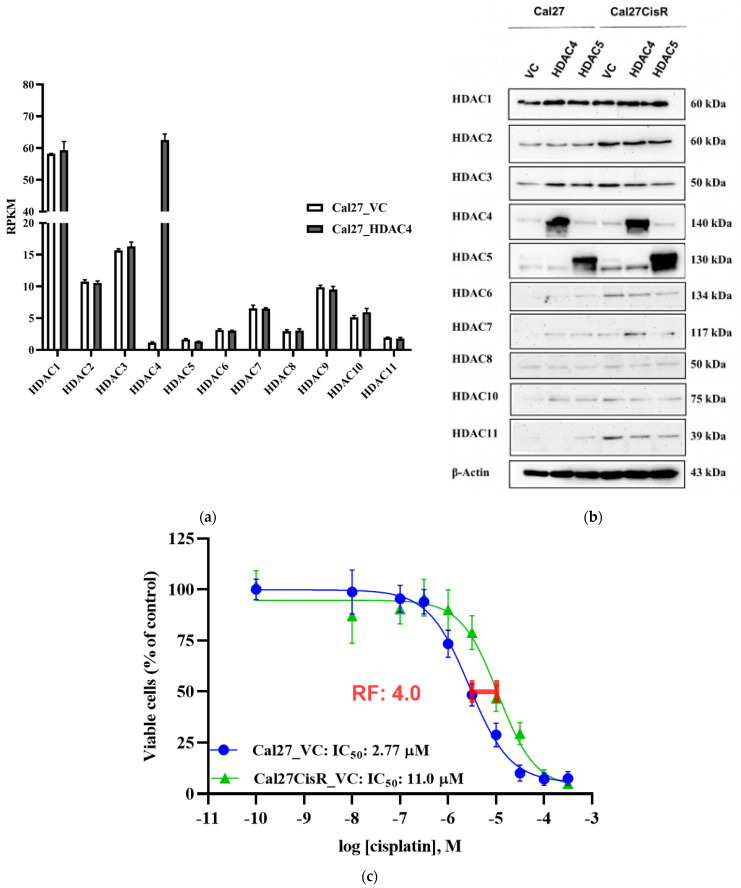
(**a**) HDAC isoform gene expression in Cal27_VC and Cal27_HDAC4. Data shown are RPKM (reads per kilobase per million mapped reads) expression values normalized from three independent RNA sequencing experiments. The white bars represent the vector control cell line Cal27_VC, the black bars represent the HDAC4 overexpression clone Cal27_HDAC4. (**b**) Representative Western blot analysis of HDAC protein expression in vector control (VC), HDAC4 and HDAC5 overexpression clones of sensitive Cal27 and cisplatin resistant Cal27CisR cells. (**c**) Cisplatin concentration effect curves and IC_50_ values for Cal27_VC and the corresponding cisplatin resistant subline Cal27CisR_VC measured by MTT assay. Resistance factor (RF) is 4-fold.

In conclusion, the HDAC expression profile demonstrated successful transduction of Cal27 and Cal27CisR cells with the lentiviral HDAC4 or HDAC5 overexpression constructs. Furthermore, acquired cisplatin resistance in Cal27CisR was accompanied by alterations in HDAC protein expression.

### 2.2. Effect of HDAC4 Overexpression on Cell Proliferation and Tumor Growth

HDAC4 has previously been shown to induce cell proliferation in glioma, osteosarcoma, gastric-, esophageal-, and colon cancer, while it had only modest effects in urothelial carcinoma cell lines and multiple myeloma [5,8,9,13,28,29,30]. To evaluate the effects of HDAC4 overexpression on cellular proliferation in the HNSCC line Cal27, MTT assays were performed over a period of 96 h (Figure 2). During the first 48 h after seeding, the HDAC4 clone grew similarly to the vector control cell line Cal27_VC. At 72 h and 96 h however, the HDAC4 overexpression clone Cal27_HDAC4 showed significantly increased proliferation compared to Cal27_VC. 

HDAC4-induced promotion of cell proliferation in vitro (Figure 2) prompted us to elucidate in vivo effects of CHDI0039 on the proliferation of Cal27_HDAC4 in comparison to the control cell line Cal27_VC in the chorioallantoic membrane (CAM) model [31]. After transplantation of the tumor cells and subsequent compound incubation for 7 days, tumors were removed, measured and weighed (Figure 3). Cal27_HDAC4 tumors showed a significant increase in volume compared to Cal27_VC. Tumor weight was also increased but not significant. Treatment of Cal27_VC with 5 µM CHDI0039 had no significant effects on tumor volume and weight, while it caused a significant decrease in volume and weight of Cal27_HDAC4 tumors. An amount of 5 µM CHDI0039 completely inhibited class IIa HDAC enzymes, whereas class I HDACs was unaffected [27].

### 2.3. Transcriptome Changes Induced by HDAC4 Overexpression and Chemical Inhibition of Class IIa HDACs

Next, we characterized the transcriptome changes induced by HDAC4 overexpression and by an inhibition of class IIa HDACs by CHDI0039, a selective and potent class IIa HDAC inhibitor [27]. RNA sequencing results were compared between HDAC4 overexpressing Cal27 cells (Cal27_HDAC4) and vector control Cal27 (Cal27_VC), and between 5 µM CHDI0039-treated versus untreated Cal27_HDAC4 cells. Table 1 shows the number of differentially expressed genes with a fold change cutoff of ≥2 and *p*-values of ≤0.05 for FDR (false discovery rate) and Bonferroni adjusted *p*-values. FDR and Bonferroni adjustments provided similar results, considering the number of regulated genes. Due to its more stringent nature, Bonferroni-adjusted genes were used for further analysis. HDAC4 overexpression resulted in a differential expression of 363 genes (Bonferroni adjustment, *p* ≤ 0.05; fold change ≥ 2) with 76.6% of genes downregulated. 

Treatment of Cal27_HDAC4 with 5 µM CHDI0039 for 24 h affected the expression of 141 genes, among which 61 genes were upregulated and 80 genes were downregulated in the treated compared to untreated Cal27_HDAC4 (Table 1). These 141 genes were then analyzed for their influence on the survival of head–neck cancer patients using publicly available data from the Kaplan–Meier (KM) plotter database [32]. Thirteen genes were identified whose up- or downregulation upon CHDI0039 treatment were significantly correlated with an improved survival of head and neck cancer patients (Table 2). Kaplan–Meier survival curves of these 13 genes are shown in Appendix A. 

Further analysis of these 13 genes with the GEPIA (gene expression profiling interactive analysis) HNSC (head–neck squamous cancer) database (Figure 4) was used to compare the median expression of these genes in normal tissue compared to tumor tissue [33]. Three genes, SERPINB2, TAGLN, and HIST1H2BD, showed high expression and significant differences between normal and tumor tissue in this analysis. TAGLN and HIST1H2BD were considerably stronger when expressed in tumor tissue compared to healthy tissue. Upon CHDI0039 treatment in Cal27_HDAC4, both genes were downregulated, thus shifting expression toward normal tissue (Table 2). TAGLN (transgelin) is part of the family of actin-associated proteins functioning as actin cross-linking protein. A high expression of TAGLN was correlated with advanced TNM stage of lung adenocarcinoma [34]. HIST1H2BD (H2B Clustered Histone 5) is a nucleosomal protein involved in the regulation of transcription and DNA repair. A high expression of HIST1H2BD predicts poor prognosis in a variety of cancers, including gliomas and HNSCC [35]. In contrast, SERPINB2 showed a higher expression in healthy tissue (Figure 4) and was upregulated upon CHDI0039 treatment, again shifting the expression in the direction of normal tissue. SERPINB2 is the plasminogen-activator inhibitor 2 (PAI2). An activation of the plasminogen-activation system leads to extracellular matrix degradation, cell proliferation, and migration. The inhibition of these pathways by SERPINB2 is thus beneficial, and consequently, high SERPINB2 expression is linked with reduced tumor growth, reduced metastasis, and prolonged survival in a variety of cancers, such as pancreatic cancer and HNSCC [36]. In addition, decreased SERPINB2 expression in head and neck cancer cell lines has been associated with acquired cisplatin resistance [37].

**Figure 4 ijms-24-05553-f004:**
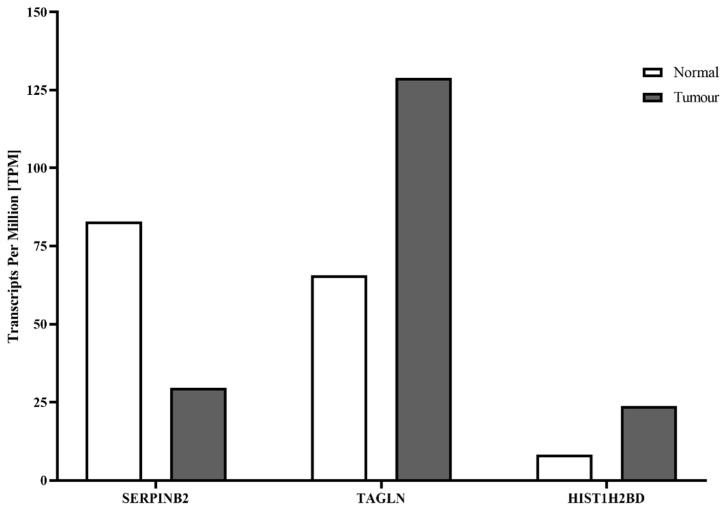
Median TPM (transcripts per million) values of SERPINB2, TAGLN, and HIST1H2BD in tumor versus normal head–neck tissue. A total of 519 HNSCC tissues and 44 normal tissues from the GEPIA database were used for this analysis. Data were taken from http://gepia.cancer-pku.cn, accessed on 12 December 2022 [33].

Taken together, our RNA expression data suggest that class IIa HDAC inhibition is beneficial for the treatment of head and neck cancers possibly by modulating the expression of genes whose up- or downregulation is known from Kaplan–Meier data to positively affect head–neck cancer patient survival. These findings prompted us to investigate the role of the class IIa HDAC inhibitor CHDI0039 together with cisplatin or bortezomib as novel treatment combinations, particularly for platinum-resistant HNSCC, as has been suggested for other tumor entities [9,10,21].

### 2.4. Effects of HDAC4 and HDAC5 on Cytotoxicity toward HDACi, Cisplatin and Bortezomib

Next, we evaluated the influence of HDAC4 and HDAC5 overexpression on the cytotoxic effect of the class IIa-selective HDACi CDHI0039, the pan-HDACi vorinostat, and the proteasome inhibitor bortezomib, as well as cisplatin, which is the standard of care for HNSCC. In general, HDAC4 and HDAC5 expression status played a minor role for the cytotoxicity of all tested compounds (Table 3). Bortezomib displayed IC_50_ values of 5 nM in the overexpression clones (HDAC4, HDAC5), respectively, and 9 nM for Cal27_VC. IC_50_ values for CHDI0039 ranged from 9.69 µM (Cal27_VC) to 14.1 µM (Cal27_HDAC5), exhibiting moderate cytotoxicity. Compared to the unselective HDACi vorinostat, which displayed IC_50_ values in the low µM range, cytotoxicity of CHDI0039 was significantly lower in all tested cell lines. As expected, cisplatin IC_50_ values were increased in the Cal27CisR clones compared to the Cal27 clones. IC_50_ values of vorinostat and CHDI0039 were also slightly increased in the Cal27CisR clones. Interestingly, IC_50_ values of bortezomib were between 2.8–4.4-fold higher in the corresponding Cal27CisR clones.

### 2.5. Effects of Class IIa HDACi on Cisplatin Induced Cytotoxicity 

An intriguing aspect of HDACi is their chemosensitizing ability when used in combination with other established anti-cancer drugs. We and others have demonstrated the importance of class I HDAC inhibition in modulating cisplatin sensitivity in a variety of different solid cancer cell lines [16,18,24,38,39]. Class IIa HDAC inhibition, HDAC4 inhibition in particular, has also been proposed to sensitize certain types of cancer to cisplatin, e.g., gastric and ovarian cancer [7,40]. To investigate whether selective class IIa HDAC inhibition can be used to sensitize HNSCC cells to cisplatin, CHDI0039 was tested in combination with cisplatin in the cisplatin-resistant Cal27CisR clones Cal27CisR_VC, Cal27CisR_HDAC4, and Cals27CisR_HDAC5 by MTT assays. Vorinostat served as control. The results are shown in Table 4. The combination was performed as a 48 h preincubation with the HDACi (or medium control) followed by the addition of cisplatin for an additional 72 h. Due to the increased total cell incubation time in this combination experiment (120 h) compared to the experiment shown in Table 3 (only 72 h), cisplatin IC_50_ values for the Cal27CisR_VC clone differ in Table 3 and Table 4 by a factor of 3.6–4.7. Overexpression of HDAC4 or HDAC5 had no significant influence on the cisplatin IC_50_ (Table 4). While 2.5 µM CHDI0039 had no impact on the cisplatin IC_50_ in any of the three cell lines, 5 µM CHDI0039 slightly reduced the cisplatin IC_50_ of Cal27CisR_VC from 51.3 µM to 32.5 µM (Table 4). Interestingly, the cisplatin IC_50_ of the HDAC4 overexpression clone increased upon treatment with 5 µM CHDI0039 from 49.9 µM to 88.7 µM, while the cisplatin IC_50_ of the HDAC5 overexpression clone showed no significant change. As shown in previous work by our group, vorinostat was able to sensitize Cal27CisR significantly to cisplatin with a shift factor (SF) of 8.9 [39]. In summary, in contrast to class I or pan-HDAC inhibition (vorinostat), class IIa inhibition by CHDI0039 did not significantly increase chemosensitivity against cisplatin in Cal27CisR, not even in HDAC4 or HDAC5 overexpressing cells. 

### 2.6. Combining Class IIa HDACi with Bortezomib Provides Synergistic Cytotoxicity in Head and Neck Cancer 

We recently showed that class IIa HDAC inhibition is synergistic with bortezomib in leukemia cell lines [21]. In multiple myeloma and pancreatic cancer, the combination of the class IIa selective HDACi TMP269 in combination with proteasome inhibitors enhanced ER-stress-mediated cell death and inhibited cell growth [9,10]. Therefore, class IIa HDAC inhibition may be a novel approach to increase the anti-cancer effects of proteasome inhibitors, which have been of limited use in solid cancers so far, mostly due to toxicity [41]. Of particular interest is if class IIa HDAC inhibition increases the potency of proteasome inhibitors to reduce clinically known toxicity. In the present study, we investigated the more potent and selective class IIa HDACi CHDI0039 (compared to TMP269) in combination with the proteasome inhibitor bortezomib by MTT assay. Bortezomib was 2–4-fold less potent in Cal27CisR clones than the respective Cal27 clones. We thus chose Cal27CisR clones to test for an increase in potency of bortezomib by a combination with the class IIa HDACi CHDI0039. Table 5 shows the IC_50_ values of bortezomib in the absence and presence of increasing concentrations of CHDI0039 in the various Cal27CisR clones. Bortezomib IC_50_ values in the absence of CHDI0039 ranged from 20.4 nM (Cal27CisR_HDAC4) to 25.3 nM (Cal27CisR_Vc) (Table 5). 

Our results show that 5 µM, 7.5 µM, and 10 µM CHDI0039 increased the cytotoxic effect of bortezomib in all tested cell lines with shift factors ranging between 2.46 to 2.98. Interestingly, 2.5 µM CHDI0039 significantly increased the cytotoxic effect of bortezomib in Cal27CisR_HDAC4 and Cal27CisR_HDAC5, while no significant change was observed for Cal27CisR_VC. Next, we examined if the combination of CHDI0039 and bortezomib is synergistic according to Chou–Talalay (Table 6). A combination index (CI) below 0.9 means that the combination is synergistic [42]. An amount of 2.5 µM CHDI0039 with 10 nM bortezomib was synergistic in HDAC4 and in HDAC5 overexpressing cells, while the fraction affected was less than 20% (*) for the vector control clone. Higher concentrations of CHDI0039 and bortezomib were synergistic in all tested cell lines, independently of HDAC4 and HDAC5 expression status. 

### 2.7. CHDI0039 and Bortezomib Synergistically Increase Caspase 3/7 Activation

CHDI0039 is synergistic with bortezomib as shown by Chou–Talalay analysis (MTT assays) in chapter 2.6. To confirm the supposed apoptotic cell death by bortezomib and CHDI0039, we investigated the activation of caspases 3/7 using fluorescence-based imaging (Figure 5). An amount of 10 µM CHDI0039 alone yielded minimal caspase 3/7 activity, irrespective of HDAC4 and HDAC5 expression status (Figure 5a–c). Using staurosporin as positive control and set as 100% caspase 3/7 activation (not shown), CHDI0039 alone and up to 10 µM resulted in only a 4–8% caspase activation (Figure 5d, striped bars). An amount of 10 nM Bortezomib alone had a slightly stronger effect on caspase 3/7 activation than CHDI0039, with the lowest effect in Cal27CisR_VC (7%) and strongest effect in Cal27CisR_HDAC4 (22%) (Figure 5a–d, striped bars). However, combined treatment with CHDI0039 and bortezomib increased caspase 3/7 activation substantially, especially when using CHDI0039 in concentrations of 7.5 µM and 10 µM (Figure 5a–c). White bars in Figure 5d show the effect of the combined drug treatments, whereas black bars show the sum of the single treatments (effect of bortezomib alone plus effect of CHDI0039 alone). Combined treatment with 10 µM CHDI0039 and bortezomib (white bars) enhanced caspase 3/7 activation by up to 52%, 73%, and 83% compared to the sum of the single treatments (black bars) in Cal27CisR_VC, Cal27CisR_HDAC4, and Cal27CisR_HDAC5, respectively (Figure 5d). Thus, caspase activation data confirm results from the MTT-based Chou–Talalay analysis (Table 6) and support a synergistic activity of CHDI0039 and bortezomib. Of note, the synergistic effect is clearly stronger in HDAC4 and HDAC5 overexpressing cells compared to the vector control (VC) clone. 

## 3. Discussion

While there are numerous studies regarding the role of class I and class IIb HDACs in tumorigenesis and chemoresistance, the function of class IIa HDACs in these processes is less well understood. HDAC4 and/or HDAC5 have been linked to tumorigenesis, tumor progression or chemoresistance in a variety of cancers, such as colon, urothelial, gastric, haematological, pancreatic, esophageal, or glioma [1,3,4,5,6,7,8,9,10,11,12,13]. HDAC4 promotes proliferation and invasion and is considered as a negative prognostic marker in regard to tumor grade and survival. For HNSCC, similar observations are published. Cheng et al. found a correlation of poor overall survival and poor progression-free survival with high HDAC4 expression in HNSCC [26]. Furthermore, loss of HDAC4 (by siRNA) sensitized TRAIL-resistant (tumor necrosis factor-related apoptosis-inducing ligand) HNSCC cells to apoptotic cell death [25]. Additionally, HDAC4 expression was found to be elevated in HNSCC compared to normal tissue from the same patients [25]. 

We found that HDAC4 overexpression by lentiviral transduction increased proliferation of the head and neck cancer cell line Cal27 significantly compared to vector control cells (Figure 2). The same result was obtained in the in vivo CAM model: HDAC4 overexpressing cells showed significantly increased tumor volume compared to vector control cells (Figure 3a) and increased, but not significantly different, tumor weight (Figure 3b). These results from head–neck cancer cells confirm the literature-known pro-tumorigenic effects of HDAC4 and support the hypothesis that class IIa HDACi are beneficial for the treatment of HDAC4 expressing cancers. 

This could be further demonstrated in the in vivo CAM model where treatment with 5 µM CHDI0039 resulted in a significant decrease in size and weight of Cal27_HDAC4 but not of Cal27_VC tumors in the CAM assay (Figure 3). These results are not a contrast to cytotoxicity data from CHDI0039 (MTT assay, Table 3), where CHDI0039 has almost the same IC_50_ for low and high HDAC4-expressing Cal27 (around 10–13 µM for Cal27_VC and Cal27_HDAC4). While we measured cytotoxicity of CHDI0039 in Table 3, the concentration of 5 µM CHDI0039 in the CAM-model—clearly below the cytotoxicity IC50 of 10–13 µM—fully inhibited class IIa HDAC enzymes according to enzyme inhibition data [27] with no inhibition of class I HDAC enzymes. In the CAM model, we then observed a decreased proliferation of Cal27_HDAC4 cells, but not of Cal27_VC. The explanation for this observation is class IIa HDAC inhibition by 5 µM CHDI0039 and not a direct cytotoxic effect, thus explaining the apparent difference between in vitro data from Table 3 and the CAM data from Figure 3. In conclusion, these results demonstrate an in vivo antiproliferative effect of CHDI0039 in Cal27 HNSCC cells displaying high HDAC4 expression. 

Additional support for a beneficial effect of class IIa HDACi in the management of HDAC4-expressing head and neck cancer cells comes from our gene expression study (RNAseq) in control and HDAC4-overexpressing Cal27 cells without and with 5 µM CHDI0039 treatment. Among differentially expressed genes (Bonferroni *p*-value ≤ 0.05; fold change ≥ 2), we identified five genes whose upregulation and eight genes whose downregulation upon CHDI treatment is correlated with a significantly increased survival of head–neck cancer patients, as found by analysis in the publicly available KM plotter database (https://kmplott.com, accessed on 15 October 2022) and displayed in Table 3 and Appendix A [32]. 

Western blot analysis revealed that class I HDACs were the most abundantly expressed isozymes in Cal27 and Cal27CisR cells (Figure 1). Results of clinical trials with HDACi as single agents especially in solid cancers have been rather disappointing [17]. It has however been suggested that the combination of (selective) HDACi with other anti-cancer-drugs is beneficial [20]. Class I selective HDACi, such as entinostat, or pan-HDACi, such as vorinostat, have been demonstrated by our group and others to sensitize cancer cells (e.g., ovarian, head and neck, small cell lung cancer) to cisplatin, a standard chemotherapeutic drug in the treatment of head and neck cancer [16,18,23,24,39,43]. However, pan-HDACi are known to be rather cytotoxic [1,20]. 

An important characteristic of class IIa HDACi is their lower cytotoxicity compared to class I- and pan-HDACi [21]. In addition, class IIa HDACi are less toxic in non-cancer than in cancer cells as shown for leukemias [21]. Our results from MTT assay further demonstrate lower cytotoxicity of CHDI0039 compared to the class I/IIb HDACi vorinostat (Table 3). IC_50_ values of CHDI0039 in Cal27 (9–14 µM, Table 4) and in Cal27CisR (22–25 µM, Table 4) were in a concentration range where this compound loses class IIa HDAC selectivity (IC_50_ HDAC4: 0.1 µM; HDAC6: 6 µM; HDAC8: 25 µM [27]), suggesting that cytotoxicity of CHDI0039 is in part due to class I/IIb inhibition. 

In accordance with our previous results, cisplatin-resistant Cal27CisR showed a strong enhancement of cisplatin cytotoxicity in combination with vorinostat (SF 16–18, Table 5). Synergistic effects of HDACi with cisplatin have been mainly attributed to the inhibition of class I HDACs, enzymes critically involved in the regulation of cell cycle, apoptosis, DNA-damage response, autophagy, and other cellular processes [12,16]. HDAC4 has been recently described to contribute to cisplatin resistance in gastric cancer [7]. HDAC4 overexpressing gastric cancers were less sensitive to cisplatin. Inhibition of HDAC4 led to enhanced cisplatin-mediated cytotoxicity, promoting apoptosis via caspase 3 activation. Moreover, HDAC4 expression was elevated in gastric tumor samples compared to healthy tissue [7]. Further, in our study, treatment of Cal27_HDAC4 cells with CHDI0039 resulted in a significant upregulation of SERPINB2 compared to untreated Cal27_HDAC4 (Table 3a). Huang et al. showed that SERPINB2 plays a role in acquired cisplatin resistance in head and neck cancers, and that upregulation of SERPINB2 can enhance sensitivity to cisplatin [37]. Therefore, we initially anticipated that the combination of a class IIa selective HDACi with cisplatin would be beneficial in the treatment of head and neck cancer cells with high HDAC4 expression. In contrast to this initial assumption, a combination of CHDI0039 with cisplatin had only minor effects on the cisplatin IC_50_ irrespective of HDAC4 and HDAC5 expression. Furthermore, neither overexpression of HDAC4 nor HDAC5 changed the cisplatin IC_50_ in Cal27 or Cal27CisR clones compared to vector control cell lines. Taken together, neither HDAC4 nor HDAC5 expression status nor class IIa HDAC inhibition had a notable influence on cisplatin sensitivity in the head and neck cancer cell lines Cal27 and Cal27CisR (shift factors < 2) in contrast to results with vorinostat (Table 5). 

However, resistance to cisplatin is clinically relevant and problematic in HNSCC patients. Mostly, palliative treatment options remain. Since the sensitization of cisplatin-resistant Cal27CisR cells against cisplatin by CHDI0039 performed unsatisfactorily, we were searching for alternative chemotherapeutics whose activity might be increased by a combination with the class IIa HDACi CHDI0039. Previous studies in pancreatic cancer, multiple myeloma, and our study on leukemia reported that dual inhibition of class IIa HDACs and the proteasome increases anti-tumor effects of bortezomib and enhances apoptosis induction [9,10,21]. In accordance with these studies, we also found a synergistic interaction of the class IIa HDACi CHDI0039 with the proteasome inhibitor bortezomib in Cal27CisR (Table 6). A combination treatment led to a reduction in cell viability and induction of apoptosis (caspase activation). While higher concentrations of CHDI0039 were synergistic in all clones, irrespective of HDAC4 and HDAC5 expression status, lower concentrations of CHDI0039 (2.5 µM) were only synergistic in cell lines with high HDAC4 or HDAC5 expression (Table 6). Synergism with higher concentrations of CHDI0039 (>5 µM) might additionally involve HDAC6 inhibition (HDAC6 enyzme IC_50_: 6 µM [27]) which has been reported to synergize with proteasome inhibitors by increasing the accumulation of misfolded proteins and enhancing ER-stress-mediated apoptosis [44]. Notably, as well as in caspase activation experiments (Figure 5), the synergistic effect is stronger in HDAC4 and HDAC5 overexpressing cells compared to control. The combination of a class IIa HDACi and proteasome inhibitor is not necessary in platinum-sensitive HNSCCs, but if platinum resistance occurs, then the synergistic combination of class IIa HDACi with proteasome inhibition may be valuable. Taken together, our results suggest that in case of cisplatin-resistant HNSCC cells, herein shown for Cal27CisR, a combination of a class IIa HDACi and a proteasome inhibitor results in a therapeutic option. 

## 4. Materials and Methods

### 4.1. Reagents

Cisplatin was purchased from Sigma-Aldrich (Steinheim, Germany) and dissolved in 0.9% sodium chloride solution. Vorinostat, CHDI00390576, and Bortezomib were purchased from Tocris (Bristol, UK). Puromycin was purchased from Invitrogen (Carlsbad, CA, USA). Stock solutions (10 mM) of the respective compounds were prepared with DMSO and diluted to the desired concentrations with the appropriate media. All other reagents were supplied by PAN Biotech (Aidenbach, Germany) unless otherwise stated.

### 4.2. Cell Culture

The human tongue cell line Cal27 was obtained from the German Collection of Microorganisms and Cell Cultures (DSMZ, Braunschweig, Germany). The corresponding cisplatin-resistant (CisR) cell line Cal27CisR was generated by exposing the parental cell line to weekly cycles of an IC_50_ of cisplatin over a period of 24–30 weeks as described in Gosepath et al. [22]. Cell lines Cal27 and Cal27CisR were grown at 37 °C under humidified air supplemented with 5% CO_2_ in DMEM containing 10% heat-inactivated fetal calf serum, 120 IU/mL penicillin, and 120 µg/mL streptomycin. The cells were grown to 80% confluency before being used in further assays.

### 4.3. MTT Cell Viability Assay

The rate of cell-survival under the action of test substances was evaluated by an improved MTT assay as previously described [45]. In brief, cells were seeded at a density of 2500 cells/well in 96-well plates (Corning, NY, USA). After 24 h, cells were exposed to increasing concentrations of the test compounds. Incubation was terminated after 72 h and cell viability was determined by the addition of MTT solution (Serva, Heidelberg, Germany, 5 mg/mL in phosphate buffered saline). The formazan precipitate was dissolved in DMSO (VWR, Darmstadt, Germany). Absorbance was measured at 544 nm and 690 nm in a FLUOstar microplate-reader (BMG LabTech, Offenburg, Germany). To investigate the effect of CHDI0039 and vorinostat on cisplatin-induced cytotoxicity, compounds were added 48 h before cisplatin administration. After 72 h, the cytotoxic effect was determined by MTT assay and shift factors were calculated by dividing the IC_50_ value of cisplatin alone by the IC_50_ value of the drug combinations. To investigate the effect of CHDI0039 on bortezomib-induced cytotoxicity, compounds were added simultaneously and incubated for 72 h before the addition of MTT solution. For proliferation assays, 2500 cells per well were seeded in 96-well plates. 

### 4.4. Synergy Studies According to Chou Talalay

To determine the synergistic effect of CHDI0039 and bortezomib, inhibition of cell viability was estimated by MTT assay. The combination indices (CIs) were calculated using Compusyn software version 1.0 (ComboSyn, Inc., Paramus, NJ, USA) based on the Chou–Talalay method [42]. 

### 4.5. Caspase 3/7 Activation Assay

Compound-induced activation of caspases 3 and 7 was analyzed by using the CellEvent Caspase 3/7 green detection reagent (Invitrogen, Carlsbad, CA, USA) according to the manufacturer’s instructions. Briefly, Cal27, and Cal27CisR clones were seeded in 96-well plates (Corning, NY, USA) at a density of 4000 cells per well. Cells were treated 24 h after seeding with indicated concentrations of CHDI0039 and bortezomib. After a further incubation period of 24 h, medium was removed and 50 µL of CellEvent Caspase 3/7 green detection reagent (2 µM in PBS supplemented with 5% heat inactivated FBS) was added. Cells were incubated for 30 min at 37 °C in a humidified incubator before imaging by using the Thermo Fisher ArrayScan XTI high content screening (HCS) system (Thermo Fisher, Waltham, MA, USA). An amount of 8µM Hoechst 33342 (Sigma-Aldrich, Steinheim, Germany) was used for nuclei staining. 

### 4.6. Generation of Cell Lines Stably Expressing HDAC4, HDAC5 or Vector Control

HDAC4 cDNA from the pcDNA-HDAC4-FLAG plasmid was kindly provided by Tso-Pang Yao (Addgene, Watertown, MA, USA, plasmid #30485). The plasmid pcDNA3.1  +  HDAC5-FLAG was a gift from Eric Verdin (Addgene plasmid #13822). Vectors puc2CL12IPwo-HDAC4-FLAG and puc2CL12IPwo-HDAC5-FLAG were generated as previously described [5,6]. Production of the lentivirus and cell transduction were performed as previously described. Briefly, production of the lentivirus was performed by transfection of HEK293-T cells with pCD/NL-BH helper plasmid expression construct, pczVSV-G envelope vector and puc2CL12IPwo, puc2CL12IPwo-HDAC4-FLAG, or puc2CL12IPwo-HDAC5-FLAG plasmid. Replication-deficient lentiviral particles were harvested 48 h after transfection using 8 µg/mL polybrene (Sigma-Aldrich, Steinheim, Germany) and were then used for transduction of Cal27 and Cal27CisR cells. The remaining viral particles in the supernatant were removed 24 h after the transduction. Selection was performed by 1 µg/mL puromycin (Invitrogen, Carlsbad, CA, USA). 

### 4.7. Chicken Chorioallantoic Membrane (CAM) Model

CAM xenografts were performed as previously described [31]. For the CAM xenografts, specific-pathogen-free chicken eggs supplied by VALO BioMedia GmbH, Osterholz-Scharmbeck, Germany, were used. Briefly, cell lines Cal27_LV and Cal27_HDAC4 were washed with PBS following trypsinization. Cells where then resuspended in PBS, counted and placed on ice until used. One million cells of either Cal27_LV or Cal27_HDAC4 were seeded on E10 CAMs and tumors were grown for 7 days. At days 2 and 4, CHDI0039 was injected below the CAM. 

### 4.8. RNA Extraction and RNA Sequencing Analysis

Total RNA was isolated using RNeasy Mini Kit (QIAGEN, Hilden, Germany) according to the instruction of the manufacturer. The RNA samples used for the RNA sequence analysis were quantified (Qubit RNA HS Assay, Thermo Fisher Scientific, Waltham, MA, USA) and quality-measured by capillary electrophoresis using the Fragment Analyzer and the ‘Total RNA Standard Sensitivity Assay’ (Agilent Technologies, Inc. Santa Clara, CA, USA). All samples in this study showed high RNA quality numbers (RQN = 10). The library preparation was performed according to the manufacturer’s protocol using the ‘VAHTS™ Universal V6 RNA-Seq Library Prep Kit for Illumina^®^ with mRNA capture module’. Briefly, 500 ng total RNA was used for mRNA capturing, fragmentation, the synthesis of cDNA, adapter ligation, and library amplification. Bead-purified libraries were normalized and finally sequenced on the HiSeq 3000/4000 system (Illumina Inc. San Diego, CA, USA) with a read setup of 1 × 150 bp. The bcl2fastq tool (v2.20.0.422) was used to convert the bcl files to fastq files as well as for adapter trimming and demultiplexing.

### 4.9. Analysis of RNA Sequencing Data 

Data analyses on fastq files were performed with CLC Genomics Workbench (version 20.0.3, QIAGEN, Venlo, NL, The Netherlands). The reads of all probes were adapter trimmed (Illumina TruSeq) and quality trimmed (using the default parameters: bases below Q13 were trimmed from the end of the reads, with a maximum of two ambiguous nucleotides). Mapping was against the Homo sapiens (GRCh38.88) (Mai 25, 2017) genome sequence. After grouping the samples (three biological replicates each) according to their respective experimental condition, the statistical differential expression was determined using the CLC—Differential Expression for RNA-Seq tool (version 2.4). The resulting *p* values were corrected for multiple testing by FDR and Bonferroni-correction. A *p* value of ≤0.05 was considered significant. Data were further evaluated with the Ingenuity-Pathway analysis software (QIAGEN, Hilden, Germany, 2016) and RaNA-Seq [46].

### 4.10. Immunoblotting

Cell pellets were dissolved with RIPA buffer (50 mM Tris-HCl pH 8.0, 1% Triton X-100, 0.5% sodium deoxycholate, 0.1% SDS, 150 mM sodium chloride, 2 mM EDTA, supplemented with protease and phosphatase inhibitors (Pierce™ protease and phosphatase inhibitor mini tablets, Thermo Scientific, Waltham, MA, USA) and clarified by centrifugation. Equal amounts of total protein (30 µg) were resolved by SDS-PAGE and transferred to polyvinylidene fluoride membranes (Merck Millipore, Burlington, MA, USA). Blots were incubated with primary antibodies against HDAC1 (Cat. No. NB100-56340), HDAC2 (Cat. No. MAB7679), HDAC3 (Cat. No. NB100-1669), HDAC4 (Cat. No. AF6205), HDAC5 (Cat. No. NBP2-22152), HDAC6 (Cat. No. NB100-56343), HDAC7 (), HDAC8 (), HDAC10 (Cat. No. NB100-91801), and HDAC11 (Cat. No. NBP2-16789). Immunoreactive proteins were visualized using luminol reagent (Santa Cruz Biotechnology, Heidelberg, Germany) with an Intas Imager (Intas, Göttingen, Germany). Relative protein quantification was performed with ImageJ software (National Institutes of Health, MD, USA) according to an internal protocol from H. Davarinejad (Davarinejad, H. Quantifications of Western Blots with ImageJ. Available online: www.yorku.ca/yisheng/Internal/Protocols/ImageJ.pdf, accessed on 26 February 2023). Briefly, the pixel density of target proteins, loading control β-actin, and their backgrounds were inverted and measured. Net pixel density for the target proteins and β-actin was then calculated by deducting the inverted background values from the inverted band values. Lastly, the net value of the target protein was divided by the net value of β-actin, resulting in the ratio of target protein/β-actin. 

### 4.11. Data Analysis

Concentration-effect curves were constructed using Prism 7 (GraphPad, San Diego, CA, USA) by fitting data of at least three experiments performed in triplicate to the four parameter logistic equation. Statistical analysis was performed using *t* test.

## 5. Conclusions

We demonstrated that the combination of class IIa HDACi and proteasome inhibitors is synergistic in cisplatin-resistant head and neck cancer cells and may constitute an alternative treatment in platin-resistant cancer. Synergy is particularly evident in HDAC4 overexpressing cell lines. Further, CHDI0039 mono-treatment reduces tumor mass and weight in high HDAC4-expressing HNSCC in vivo (CAM model). Noteworthy is the much lower cytotoxicity of class IIa HDACi compared to class I- or pan-HDACi. In conclusion, class IIa HDACi—firstly—reduce tumor growth of HNSCC with high HDAC4 expression and—secondly—act synergistically in combination with proteasome inhibitors in platinum-resistant HNSCC. HDAC4 expression may be useful as a (predictive) biomarker for the use of class IIa HDACi.

## Figures and Tables

**Figure 2 ijms-24-05553-f002:**
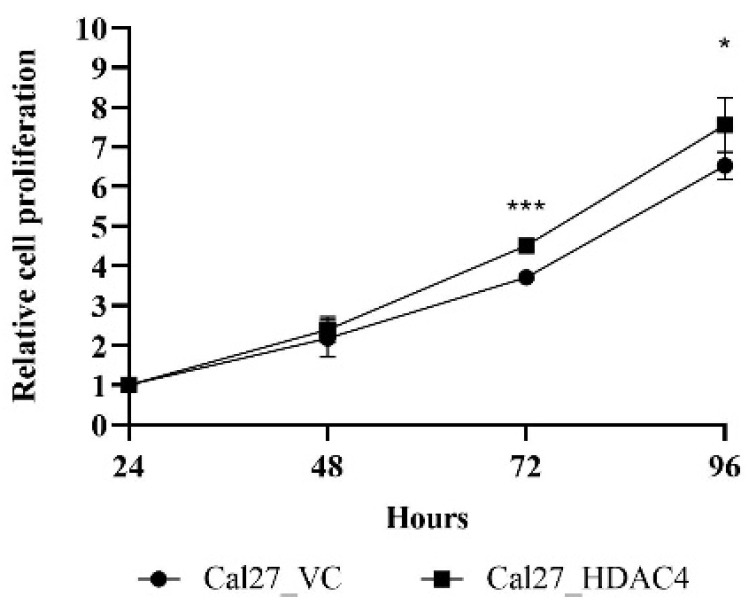
Effects of HDAC4 overexpression on cell proliferation analyzed over a time period up to 96 h by MTT assay. The value for 24 h was defined as a starting point with a relative proliferation set as 1. Values represent means ± SD of at least three independent experiments. Significance was calculated by *t*-test (* *p* ≤ 0.05, *** *p* ≤ 0.001).

**Figure 3 ijms-24-05553-f003:**
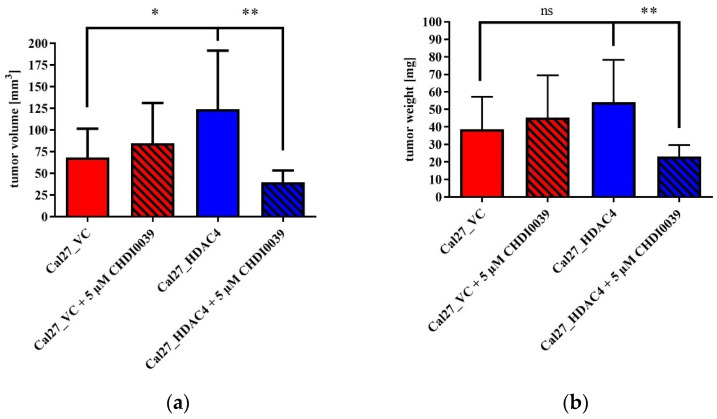
Effects of HDAC4 overexpression and treatment with 5 µM CHDI0039 on tumor volume (**a**) and weight (**b**) in the chorioallantoic membrane (CAM) model. Tumors were seeded on the CAM and grown for 7 days. Treatment with CHDI0039 or buffer control was conducted at day 2 and day 4 after seeding. Values represent means ± SD of at least three independent experiments. Significance was calculated by *t*-test (ns: not significant; * *p* ≤ 0.05; ** *p* ≤ 0.01).

**Figure 5 ijms-24-05553-f005:**
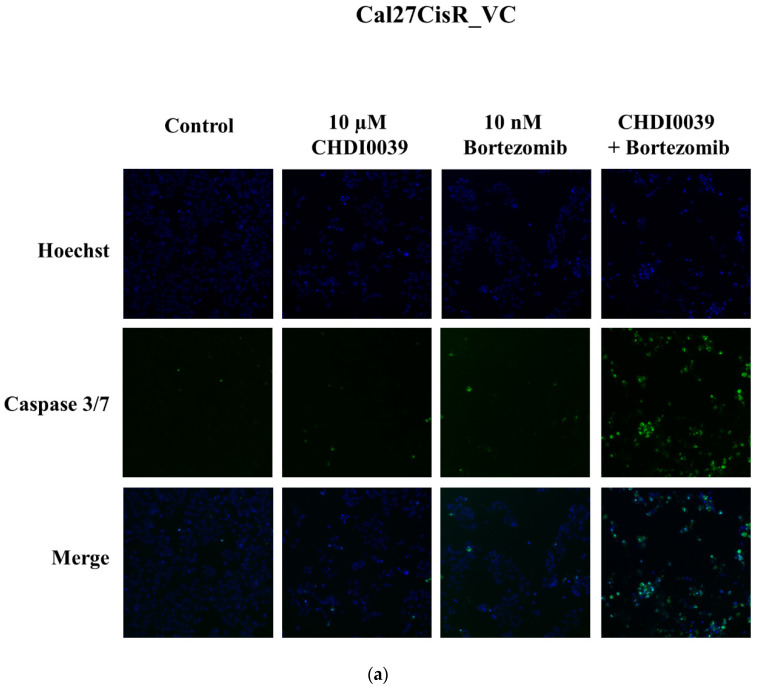
Representative fluorescent images of various Cal27CisR cell clones either untreated (control) or treated with 10 µM CHDI0039, or 10 nM bortezomib, or the combination thereof for 24 h. Hoechst 33342 (“Hoechst”, blue color) was used as nuclear staining and CellEvent Caspase-3/7 green detection reagent (green color) was used for caspase 3/7 activation. (**a**) Cal27CisR_VC cells. (**b**) Cal27CisR_HDAC4 cells. (**c**) Cal27CisR_HDAC5 cells. (**d**) Caspase 3/7 activation by single treatment and combinations of different concentrations of CHDI0039 (CHDI) and 10 nM bortezomib (Bort) in Cal27CisR_VC, Cal27CisR_HDAC4, and Cal27CisR_HDAC5. Cells were incubated with CHDI0039, bortezomib, or a combination thereof for 24 h. An amount of 0.5 µM Staurosporin was incubated for 8 h and served as positive control. Caspase 3/7 activation was analyzed by ArrayScan XTI. Data are the mean ± SD of two experiments, each with three replicates. Statistical analysis to compare the caspase 3/7 activation of the indicated treatments was performed using *t*-test. ns *p* > 0.05, * (*p* ≤ 0.05), ** (*p* ≤ 0.01), *** (*p* ≤ 0.001), **** (*p* ≤ 0.0001).

**Table 1 ijms-24-05553-t001:** Differentially expressed genes associated with HDAC4 expression status and class IIa HDAC inhibition by 5 µM CHDI0039. Shown are the total number of differentially expressed genes (upper number) and the number of upregulated (up)- or downregulated (down) genes.

Cell Line Cal27	No. Genes FDR ^1^: *p* ≤ 0.05 Fold Change ≥ 2	No. Genes Bonferroni: *p* ≤ 0.05 Fold Change ≥ 2
Cal27_HDAC4 compared to Cal27_VC	Total: 588 167 up in HDAC4 421 down in HDAC4	Total: 363 85 up in HDAC4 278 down in HDAC4
Cal27_HDAC4 + 5 µM CHDI0039 compared to Cal27_HDAC4	Total: 319 133 up in HDAC4 + CHDI0039 186 down in HDAC4 + CHDI0039	Total: 141 61 up in HDAC4 + CHDI0039 80 down in HDAC4 + CHDI0039

^1^ FDR: false discovery rate.

**Table 2 ijms-24-05553-t002:** Genes whose up- (**a**) or down- (**b**) regulation upon treatment with 5 µM CHDI0039 for 24 h in Cal27_HDAC4 promote increased survival in head–neck cancer patients according to analysis by KM plotter database [32].

(a)				
No.	Gene ID	Bonferroni from RNAseq	Fold Difference	Log Rank *p*-Value from KM Plotter
1	SERPINB2	1.16 × 10^−65^	3.47	0.035
2	NFATC2	1.15 × 10^−23^	3.38	0.013
3	PLA2G4D	9.68 × 10^−7^	2.66	0.037
4	CYTH4	1.87 × 10^−3^	2.56	0.027
5	L3MBTL1	1.31 × 10^−2^	2.11	0.016
**(b)**				
**No.**	**Gene ID**	**Bonferroni** **from RNAseq**	**Fold Difference**	**Log Rank** ***p*-Value from KM Plotter**
1	IGSF1	3.02 × 10^−2^	−2.02	0.021
2	HIST1H2BD	2.07 × 10^−7^	−2.10	0.028
3	ACTBL2	3.31 × 10^−4^	−2.13	0.027
4	MSRB3	3.43 × 10^−14^	−2.20	0.020
5	PIGR	7.53 × 10^−27^	−3.10	0.045
6	TAGLN	7.59 × 10^−10^	−3.93	0.010
7	LRRC4	3.39 × 10^−7^	−4.03	0.001
8	CAMK2A	6.92 × 10^−7^	−4.24	0.001

**Table 3 ijms-24-05553-t003:** Cytotoxic activity (MTT assay) of vorinostat, CHDI0039, bortezomib, and cisplatin at Cal27 and Cal27CisR vector control (VC) and HDAC4 and HDAC5 overexpression clones after compound incubation for 72 h. Shown are IC_50_ values ± SD of at least three independent experiments.

Compound	Cal27 Clones			Cal27CisR Clones		
	VC	HDAC4	HDAC5	VC	HDAC4	HDAC5
Cisplatin [µM]	2.77 ± 0.20	2.47 ± 0.20	3.28 ± 0.21	11.0 ± 1.85	13.7 ± 1.77	11.3 ± 1.41
Vorinostat [µM]	1.33 ± 0.08	1.49 ± 0.10	0.92 ± 0.06	2.12 ± 0.13	2.04 ± 0.09	2.58 ± 0.20
CHDI0039 [µM]	9.69 ± 0.62	12.8 ± 0.53	14.1 ± 0.61	22.6 ± 1.11	24.7 ± 1.39	24.5 ± 2.19
Bortezomib [µM]	0.009 ± 0.0002	0.005 ± 0.0002	0.005 ± 0.0004	0.025 ± 0.003	0.020 ± 0.003	0.022 ± 0.003

**Table 4 ijms-24-05553-t004:** IC_50_ values of cisplatin [µM] in the absence and presence of 2.5 μM/5 μM CHDI0039 or 0.75 µM vorinostat. ns (not significant), *** (*p* ≤ 0.001). SD ≤ 10% of the mean.

	Cal27CisR					
	VC		HDAC4		HDAC5	
	Cisplatin [µM]					
	IC_50_	SF	IC_50_	SF	IC_50_	SF
Control	51.3	-	49.9	-	46.1	-
+2.5 µM CHDI0039	48.9	1.05 (ns)	45.8	1.09 (ns)	54.6	0.84 (ns)
+5.0 µM CHDI0039	32.5	1.58 (***)	88.7	0.56 (***)	39.5	1.17 (ns)
+0.75 µM Vorinostat	3.18	16.1 (***)	3.01	16.6 (***)	2.60	17.7 (***)

**Table 5 ijms-24-05553-t005:** Coincubation of CHDI0039 with bortezomib sensitizes Cal27CisR cells with overexpression of HDAC4, HDAC5, or vector control (VC) to bortezomib. Shown are IC_50_ values of bortezomib [nM] and shift factors [SF]. † = *** (*p* ≤ 0.001). SD ≤ 15% of the mean. Values represent means of three independent experiments.

		Cal27CisR					
		VC		HDAC4		HDAC5	
		Bortezomib [nM]					
		IC_50_	SF	IC_50_	SF	IC_50_	SF
		25.3	-	20.4	-	21.6	-
CHDI0039 [µM]	+2.5	22.1	1.14	9.62	2.12 ^†^	11.0	1.96 ^†^
	+5.0	9.62	2.63 ^†^	7.91	2.58 ^†^	8.74	2.47 ^†^
	+7.5	8.48	2.98 ^†^	6.99	2.92 ^†^	8.03	2.69 ^†^
	+10	8.88	2.85 ^†^	7.43	2.75 ^†^	7.74	2.79 ^†^

**Table 6 ijms-24-05553-t006:** CI values according to Chou–Talalay of the drug combination CHDI0039 and bortezomib in Cal27CisR_VC, Cal27CisR_HDAC4, and Cal27CisR_HDAC5. * = fraction affected less than 20%. Values represent means of at least three independent experiments.

**(a)**				
		**Cal27CisR_VC**		
		Bortezomib [nM]		
		10	15	20
CHDI0039 [µM]	+2.5	*	0.51	0.61
	+5.0	0.69	0.54	0.68
	+7.5	0.58	0.60	0.73
	+10	0.63	0.66	0.78
**(b)**				
		**Cal27CisR_HDAC4**		
		Bortezomib [nM]		
		10	15	20
CHDI0039 [µM]	+2.5	0.76	0.53	0.67
	+5.0	0.55	0.59	0.74
	+7.5	0.52	0.64	0.79
	+10	0.57	0.70	0.85
**(c)**				
		**Cal27CisR_HDAC5**		
		Bortezomib [nM]		
		10	15	20
CHDI0039 [µM]	+2.5	0.59	0.50	0.62
	+5.0	0.47	0.56	0.68
	+7.5	0.48	0.62	0.75
	+10	0.55	0.67	0.81

## Data Availability

The data presented in this study are available in this article and the corresponding supplemental information. In addition, data are available under https://kmplot.com, accessed on 15 October 2022, and http://gepia.cancer-pku.cn, accessed on 12 December 2022.

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
