# Peer review of "Synergistic Interaction of the Class IIa HDAC Inhibitor CHDI0039 with Bortezomib in Head and Neck Cancer Cells"

_ijms, 2023, doi:10.3390/ijms24065553_

Round 1

Reviewer 1 Report

In this manuscript, Schrenk et al. explored the role of class IIa HDACi as anti-cancer chemo-sensitizing agents in head and neck squamous cell cancer (HNSCC). They focused on investigating the effects of HDAC4 in particular and the class IIa HDACi CHDI0039 on proliferation and chemosensitivity in parental and cisplatin-resistant head and neck squamous cells. They found that overexpression of HDAC4 could promote tumor growth, and treatment with CHDI0039 resulted in a significant decrease in tumor size and weight in HDAC4 overexpression cells. They also found that the combination of CHDI0039 with bortezomib could induce synergistic cytotoxicity in HDAC4 overexpression cells. From RNAseq data, the authors found that treatment with CHDI0039 alters the expression of genes whose up- or downregulation is associated with increased survival in HNSCC patients. Their conclusion is that the combination of class IIa HDACi with proteasome inhibitors could be an effective treatment option for high HDAC4 HNSCC patients.

Although this is an interesting study, there are several questions that still need to be clarified.

1.      Several spelling mistakes were found, such as “analysed”, “dependant”, please correct it.

2.      Dose the HADC5 mRNA expression also increase in HDAC5 clone?

3.      The authors should show the quantification of protein expressions in their western blot data.

4.      Line 127 “table 1c” should be corrected to “figure 1c”, in addition, the authors should correct the unit which showed in figure 1c.

5.      In Figure 3a-c, the authors should show the correspond colors in hoechst and caspase3/7, respectively

6.      In table 3 and 4, why did the IC50 of cisplatin alone have much difference in Cal27CisR cells?

7.      CHDI0039 could not increase the cytotoxicity in Cal27_HDAC4 cell in vitro but significantly suppress the tumor growth of Cal27_HDAC4 cell in vivo. The results from in vitro study (table 3) and in vivo study (Figure 5) are conflicting. Please explain or discuss it.

Reviewer 2 Report

The authors have transfected parental parental Cal27 cells and cisplatin resistant Cal27CisR cells to overexpress either HDAC4 or HDAC5 in an attempt to demonstrate that the class IIa HDAC inhibitor CHDI0039 might be more effective. However, it doesn’t appear that HDAC4 or HDAC5 overexpression predisposes the line to be sensitive to CHDI0039. The drug also does not appear to sensitize Cal27CisR cells to cisplatin. However, marginal effects were observed when CHDI0039 was combined with bortezomib. It isn’t clear that these effects would be observed widely as only one cell type was used. The authors suggest that the CHDI0039/bortezomib combination would work in head and neck cancers with high HDAC4, but it isn’t clear if high levels are observed clinically. The following issue would need to be resolved before publication.

I have to admit, I’m not clear on the rationale of generating the cell lines that overexpress HDAC4 and HDAC5. Do head and neck cancers overexpress these proteins? Was this done in the hopes that CHDI0039 would be more effective? Additionally, it isn’t clear why the authors transfected the Cal27CisR line, as they seem to use one and then then other in various sections of the manuscript, and the lines are not systematically addressed. In Figure 1A, the authors examine Cal27_HDAC4 cells but then none of the other lines are examined. For Table 1, only Cal27_HDAC4 cells are examined. For Tables 4, 5 and 6, they used the Cal27_HDAC4 line only? Then when looking at effects of HDAC4 on proliferation in Figure 5, they go back to the Cal27_HDAC4 line. The rationale for the cell lines and the data presentation needs to be better organized. Perhaps Figures 4 and 5 should be after figure 1, since the authors are focusing mainly on the Cal27 line, and then switch to the resistant line in the last sections.

In paragraph 2 of the results section, the authors state “acquired cisplatin resistance in Cal27CisR cells was accompanied by alterations in HDAC protein expression”. Which ones? They all look comparable in Figure 1B. Maybe HDAC2 is slightly higher, but actin levels look slightly higher in the Cal27CisR cells and might explain the higher HDAC2 levels. Are the differences significant?

In section 2.2, why were only HDAC4 cells used and not HDAC5? Why was 5 µM CHDI0039 used—is this a known concentration to inhibit HDACs? What are the downstream effects of this treatment (i.e. how do the authors know this works in the cell lines). While the authors show some concordance with 3 genes from their RNA Seq analysis and clinical data from head and neck cancers, what do these genes do? Are they HDAC4/5 targets? Does overexpression of HDAC 4 or 5 correlate with poor prognosis in head and neck cancer?

In section 2.3, in table 3, the authors note a “profound effect” in the Cal27 clones with HDAC4 or 5, but a change from 9 nM to 5 nM doesn’t sound like a profound sensitivity—is this statistically significant? The authors also seem to suggest that resistance to cisplatin would predict resistance to bortezomib based on the Cal27CisR cells being slightly more resistant to bortezomib. However, this is only one cell line; more would need to be queried to arrive at such a conclusion.

In section 2.4, in table 4, the authors now have the IC50 for cisplatin in the Cal27CisR cells at 51.3-46.2 in the clones; why are they now higher when in figure 1C the IC50 was around 11 µM? Also, the IC50 for CHDI0039 in the Cal27CisR clones is around 22-24 µM. The authors are now treating with 5 µM CHDI0039 in combination with cisplatin; does CHDI0039 have no effect at all at this concentration?

In section 2.5, the authors only use Cal27_HDAC4, what about HDAC5? Why not use any of the Cal27 clones? Despite the fact that vorinostat works better than CHDI0039, the authors still decide to use CHDI0039 in further studies. Why do this when CHDI0039 is clearly inferior to vorinostat in terms of potency and efficacy?

For section 2.7, the authors go back to the Cal27 model to examine tumor growth; why is only HDAC4 examined? Why is the Cal27_HDAC4 line now sensitive to CHDI0039 in Figure 5, when it was more resistant to the drug in Table 3 and why is Cal_27 VC now apparently resistant?

From their studies, the authors have concluded—as noted at the end of the abstract—that combination of class IIa inhibitors with proteasome inhibitors is effective in HNSCC cancers with high HDAC4 expression. However, the authors have only shown this in a cell line that is also resistant to cisplatin; they have not demonstrated this in the cisplatin-sensitive Cal27 line. So, theoretically, the HNSCC tumor would need to be cisplatin resistant—with the same mechanism of resistance as the Cal27CisR line—in order to be sensitive to the CHDI0039/bortezomib combination. Thus, the authors have not justified their conclusions.

Additionally, the authors have only used one cell line; more would be needed to demonstrate that this was not just unique to Cal27 cells.

Round 2

Reviewer 1 Report

In Figure 1c, the typing of "µ" is wrong, please check it.